# Tissue-specific changes in size and shape of the ligaments and tendons of the porcine knee during post-natal growth

Stephanie G. Cone[1,2], Hope E. Piercy[1], Emily P. Lambeth[1], Hongyu Ru[3], Jorge A. Piedrahita[2], Jeffrey T. Spang[4], Lynn A. Fordham[5], Matthew B. Fisher[1,2,4]*

1 Department of Biomedical Engineering, North Carolina State University and the University of North Carolina at Chapel Hill, Raleigh, North Carolina, United States of America, 2 Comparative Medicine Institute, North Carolina State University, Raleigh, North Carolina, United States of America, 3 Department of Biostatistics, North Carolina State University, Raleigh, North Carolina, United States of America, 4 Department of Orthopedics, University of North Carolina at Chapel Hill, Chapel Hill, North Carolina, United States of America, 5 Department of Radiology, University of North Carolina at Chapel Hill, Chapel Hill, North Carolina, United States of America

* mbfisher@ncsu.edu

**Data Availability Statement:** All relevant data are within the manuscript and its Supporting Information files.

## Abstract

Prior studies have analyzed growth of musculoskeletal tissues between species or across body segments; however, little research has assessed the differences in similar tissues within a single joint. Here we studied changes in the length and cross-sectional area of four ligaments and tendons, (anterior cruciate ligament, patellar tendon, medial collateral ligament, lateral collateral ligament) in the tibiofemoral joint of female Yorkshire pigs through high-field magnetic resonance imaging throughout growth. Tissue lengths increased by 4- to 5-fold from birth to late adolescence across the tissues while tissue cross-sectional area increased by 10–20-fold. The anterior cruciate ligament and lateral collateral ligament showed allometric growth favoring change in length over change in cross-sectional area while the patellar tendon and medial collateral ligament grow in an isometric manner. Additionally, changes in the length and cross-sectional area of the anterior cruciate ligament did not increase as much as in the other ligaments and tendon of interest. Overall, these findings suggest that musculoskeletal soft tissue morphometry can vary within tissues of similar structure and within a single joint during post-natal growth.

## Introduction

Joints within the musculoskeletal system consist of a complex combination of active and passive tissues including ligaments and tendons that have specific morphometric and mechanical properties enabling force transmission and movement. Many studies have investigated early pre-natal development of ligaments and tendons [1–6]. In addition, the structure, function, and biochemical makeup of ligaments and tendons undergo major changes throughout both pre-natal and post-natal growth [7–11]. Specific changes include increasing macroscale size and mechanical stiffness and changing orientation and shape, among others. These age-related

**Funding:** This work received funding from the National Institute of Arthritis and Musculoskeletal and Skin Diseases (https://www.niams.nih.gov/) of the National Institutes of Health under award numbers R03 AR068112 and R01 AR071985 to MF. The content is solely the responsibility of the authors and does not necessarily represent the official views of the National Institutes of Health. SC received funding from the National Science Foundation Graduate Research Fellowship Program (https://www.nsfgrfp.org/) under grant number DGE-1252376. Any opinions, findings, and conclusions or recommendations expressed in this materials are those of the authors and do not necessarily reflect those of the National Science Foundation. The funders had no role in study design, data collection and analysis, decision to publish, or preparation of the manuscript.

**Competing interests:** The authors have declared that no competing interests exist.

changes are influenced by a variety of stimuli including biochemical and cell signaling as well as mechanical loading [12].

Previous work by D'Arcy Thompson [13], Julian Huxley [14], and many others, have reported changes in the size and shape of biological tissues, resulting in the establishment of many terms and methods for classifying objects during growth [13–15]. In this work, we focus on the terms isometry and allometry as defined by Huxley and how they can be used to describe morphological changes in tissues during skeletal growth. The terms isometry and allometry describe changes in which the growth of a part, do or do not, match the growth of the whole, respectively [14, 15]. Further research has built on this foundation to better understand morphologic changes in the musculoskeletal soft tissues, often with a focus on differences and similarities across tissues or between species [16–18]. In this work, we apply these methods of characterization to different tissues with similar structure and function within a single organ.

Additional studies have investigated specific aspects of post-natal growth within a single tissue on the macroscale. For example, the lapine medial collateral ligament (MCL) experiences growth along the full length of the tissue, with larger increases close to the tibial insertion site [19]. Interestingly, differences in growth rate coefficients were found between the proximal bones of the hindlimb (femur) and forelimb (humerus) in the porcine model through 3 months of age but not between the distal bones of the same limbs (tibia and radius) [20]. The same study found that both the tibia and femur experienced more rapid change in bone area relative to bone length (allometric growth), although the same trend was not found in the humerus [20]. A study in human growth found that the anterior cruciate ligament (ACL) experiences linear volumetric growth up to 10 years of age, with a plateau in ACL volume during the remaining period of growth during adolescence, showing age-specific allometric growth patterns between the ACL and the body [8]. Together, these studies show that ligaments undergo changes in CSA and length during post-natal growth and that tissues near the same joint can undergo different patterns of growth. However, it is unknown if ligaments and tendons within a single joint undergo similar or different changes during post-natal growth.

The objective of this study was to analyze the post-natal morphometry of four soft tissues with similar structure and function in the same joint: the ACL, patellar tendon (PT), MCL, and lateral collateral ligament (LCL) of the knee joint. In order to address this objective, we utilized a well-described porcine model to serve as an analog for the human knee [21–23]. Magnetic resonance imaging (MRI) was performed to collect high-resolution images of joints from animals of different ages, and the macroscale size and shape of each tissue of interest was analyzed. We assessed the isometry or allometry within and between each of these tissues by comparing relative changes in tissue length and CSA over time.

## Materials and methods

### Specimen collection

Hind limbs were collected post-mortem from 36 female Yorkshire cross-breed pigs from birth to 18 months of age (n = 6/age group, total n = 36). Specific age groups and estimated human equivalent age were 0 months (newborn), 1.5 months (early juvenile), 3 months (late juvenile), 4.5 months (early adolescent), 6 months (adolescent), and 18 months (late adolescent). Human age equivalencies were based on a combination of skeletal and sexual age scales in both species [24]. The animals used in this study were obtained from a university owned herd, and all animals were healthy and of normal size. Swine were housed in barns with food and water provided according to the management practices outlined in the Guide for the Care and Use of Agricultural Animals in Teaching and Research and their use in the current experimental

protocols were approved by the N.C.S.U. Institutional Animal Care and Use Committee [25]. Animals were euthanized by trained personnel in a manner consistent with the AVMA Guidelines for the Euthanasia of Animals [26]. Hind limbs were dissected to the stifle (knee) joint and wrapped in saline-soaked gauze prior to storage at -20°C until further testing.

## Magnetic resonance imaging

Limbs were allowed to thaw at room temperature prior to imaging. All limbs were imaged while constrained to full extension (~40° of flexion in the pig) using MRI scanners at the Biomedical Research Imaging Center (BRIC) at the University of North Carolina–Chapel Hill. Due to the small size of the newborn hind limbs and the need for smaller voxel sizes than available on the primary 7.0T scanner, imaging for this group was performed using a 9.4-Tesla Bruker BioSpec 94/30 USR machine (Bruker BioSpin Corp, Billerica, MA) with a 3D fast low angle shot scan sequence (3D-FLASH, flip angle: 10°, TR: 38 ms, TE: 4.42 ms, acquisition time: 13 hours 18 minutes, FOV: 30 x 30 x 30 mm) using a 35 mm volume coil and isotropic voxels of 0.1 x 0.1 x 0.1 mm with no gap between slices. Limbs from the older age groups (1.5 to 18 months) were imaged using a 7.0-Tesla Siemens Magnetom machine (Siemens Healthineers, Erlangen, Germany) with a double echo steady state (DESS, flip angle: 25°, TR: 17 ms, TE: 6 ms, acquisition time: 24 minutes, FOV: 123 x 187 x 102 mm) using a 28 channel knee coil (Siemens Healthineers) and voxels of 0.42 x 0.42 x 0.4 mm with no gap between slices.

## Image post-processing

Image sequences were imported into commercial software (Simpleware 7.0, Synopsys, Chantilly, VA) and 3-dimensional (3D) models were generated for each tissue of interest (ACL, PT, MCL, LCL) (Fig 1). All models and measurements were generated and performed by a single author (SC), and repeatability for this process for these ligaments and tendons has been established to be highly repeatable, with intrauser correlation tests resulting in data fitting with $R^2$ values of 0.97–0.98 and interuser correlation tests resulting in data fitting with $R^2$ values of 0.84–0.99 across parameters. Models were refined using the "close" and "discrete Gaussian" filters and were exported from the software as .stl files. Models were translated into point clouds, which were then analyzed in a custom Matlab code. Comparison studies of the 3D reconstructions of *in situ* menisci calculated from these MRI scans to a gold-standard surface reconstruction based on 3D laser scans (FARO Edge ScanArm ES, Lake Mary, FL) indicated a root mean square error (RMSE) of 0.57±0.1mm for points along the object surfaces and an RMSE of 0.66 ±0.2mm for the centroid points of menisci. Any primary sources of error translating from MRI-based measurements to physical tissue morphometries should be systemic, resulting in little impact on our comparisons between age groups. Length was defined as the magnitude of the vector between the centroids of the femoral and tibial insertion sites (ACL, MCL, LCL) or the patellar and tibial insertion sites (PT). Centroids were defined as the geometric center of the points surrounding the insertion of the soft-tissue into the bone. Length measurements were complicated by insertion sites that have a substantial directional component parallel to the length of the ligament or tendon. This included the insertions of the MCL, LCL, and the tibial insertion of the PT, and some insertion sites extended beyond the field of view of the MRI scans for larger specimens. As such, the location of these insertion sites were measured at the edge of the insertion most proximal to the joint center since this landmark could be consistently identified in all specimens. The insertion was determined as the centroid of points collected around the tissue at this location along the tissue length. To compare across tissues within specific ages, values for length were normalized to the average value at 18 months for each tissue. For example, at birth, the average ACL length was 25% of the average ACL length

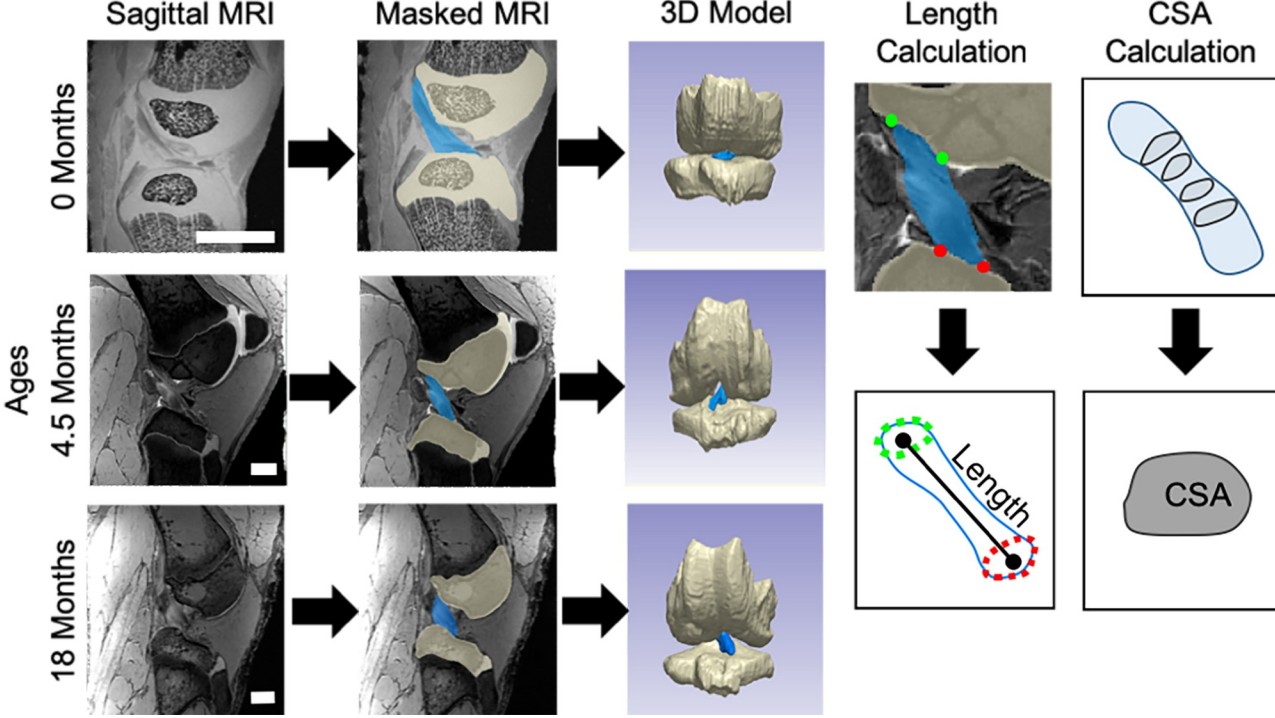

**Fig 1. 3D models were created for tissues such as the anterior cruciate ligament (ACL) using MRI scans and segmentations of individual tissues.** Images shown for newborn (0 month), early adolescent (4.5 month), and late adolescent (18 month) joints, scale bars are 10 mm. Length and cross-sectional area (CSA) calculation methods described for an ACL.

at 18 months. Furthermore, to avoid variability caused by the changing CSA near the bony insertions, our CSA analysis was restricted to the midsubstance of the tissues. Specifically, the CSA was measured from the central 50% (midsubstance) of the ligament or tendon by rotating the model of the tissue onto the longitudinal axis, dividing the model into slices at a 0.1 mm increments along the z-axis, measuring the area of each slice, and averaging the values within the central 50% along the length to collect a single value for each tissue. Values for CSA were also normalized to the average 18 month value for each tissue.

## Analysis of growth

Data were analyzed for each parameter (length and CSA) of each tissue (ACL, PT, MCL, and LCL) with comparisons performed between tissues and between parameters using data from all age groups. In order to account for non-linear tissue growth characteristics, log-log plots ($\log_{10}$) were created comparing experimental data to isometric slopes listed in Fig 2. This process was done for both intra-tissue comparisons (CSA versus length) and inter-tissue comparisons (ACL versus PT, ACL versus MCL, ACL versus LCL, PT versus MCL, PT versus LCL, MCL versus LCL) for each geometric parameter. Isometric slopes of 1 (length vs length, CSA vs CSA) or 2 (CSA vs length) were used to account for differences in dimensions between mm and mm$^2$ [17]. Linear regressions were performed for each plot and the slope and R$^2$ value were recorded.

## Statistical analysis

For comparisons between tissues, normalization of tissue size was performed by dividing the data for each geometric parameter by its respective average 18 month value for each tissue.

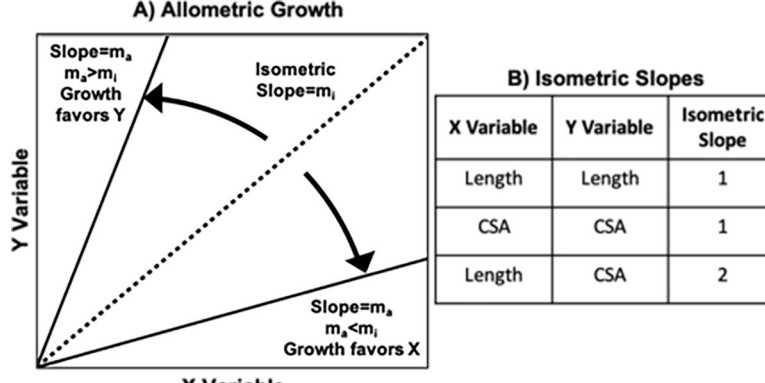

**Fig 2.** Allometric growth within or between tissues can be assessed by plotting data on a log-log graph (A) and comparing to the slope of an isometric line. Isometric slopes are listed for all possible combinations of CSA and length (B).

Normality of data sets was confirmed in JMP (JMP Pro 13, SAS Institute Inc., Cary, NC). Statistical analysis of each geometric parameter consisted of a two-way ANOVA with age and tissue type as major effects and a Bonferroni method to adjust for multiplicity and significance set at $p \leq 0.05$ (JMP Pro 13, SAS Institute Inc., Cary, NC). For these analyses, tissue type was considered a within-subject variable while age was considered a between-subject variable. Analysis of the log-log plots was accomplished by comparing the slope of the linear regression to the appropriate isometric value by an F-test by using the test statement in PROC REG Procedure (SAS 9.4, SAS Institute Inc., Cary, NC). The adjusted significance level for F-test comparisons was set at $p \leq 0.001$. Throughout the results section, data are presented as mean ± standard deviation, and 95% confidence intervals are available in the Supplemental Material.

## Results

### Changes in size during post-natal growth

All of the ligaments and tendons of interest experienced significant growth in both length (S1 Table) and CSA (S2 Table) between birth and late adolescence (18 months) in this study (Fig 3). Increasing age resulted in significant growth in all four tissues of interest ($p < 0.05$). Specifically, the length of the ACL increased 4-fold from an average of 9 ± 1 mm to 35 ± 2 mm from birth through late adolescence (Fig 3A). Simultaneously, the average length of the PT increased by 5-fold from 14 ± 1 mm to 74 ± 9 mm. The length of the MCL and the LCL also increased by 5-fold (Fig 3A). CSA increases varied across the tissue types. In the ACL, the average CSA increased 10-fold from 6 ± 1 mm$^2$ to 57 ± 9 mm$^2$ between birth and late adolescence (Fig 3B). This increase occurred alongside 24-fold (PT), 23-fold (MCL), and 16-fold (LCL) increases in the other ligaments and tendon types (Fig 3B). The most rapid periods of growth occurred during the juvenile and early adolescent phases (statistically significant increases between consecutive age groups are highlighted in Fig 3 ($p < 0.05$)).

Similar length values were obtained at birth for the PT, MCL, and LCL (19%, 20%, and 22%, respectively) (Fig 4, S3 Table). Across all ages, these average changes in normalized tissue length were only statistically significant between the ACL and the PT, MCL, and LCL at 1.5 months of age, the ACL and the MCL at 3 months of age, and the PT and the LCL at 6 months of age ($p < 0.05$).

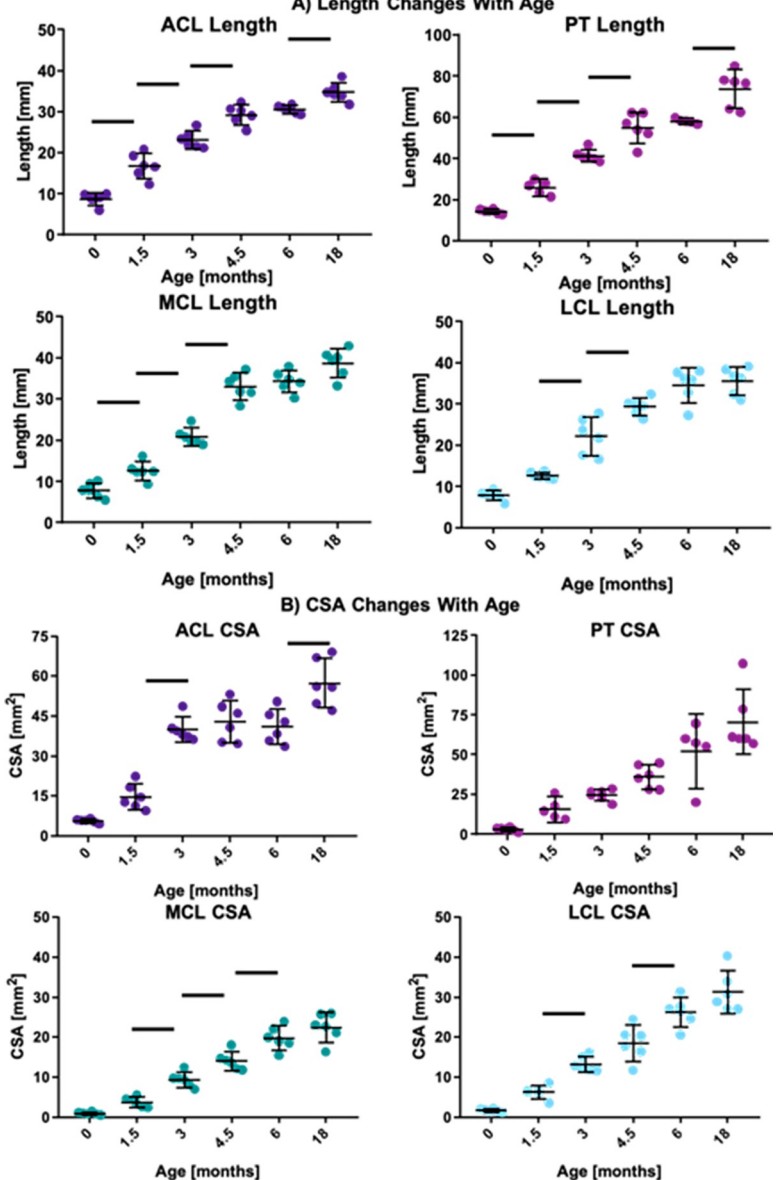

**Fig 3.** Length (A) and CSA (B) increase several fold in the ACL, PT, MCL, and LCL from birth through skeletal maturity. Data for individual specimens are presented as points while mean and 95% confidence intervals are represented by dashes and lines. Bars represent significant differences between consecutive age groups (p<0.05).

At birth, major differences in the normalized CSA of the tissues were observed (Fig 5, S4 Table). Interestingly, the newborn ACL CSA was only 10% of the average 18 month group value. This normalized CSA value was much higher compared to the other tissues of interest, as the CSA values of the PT, MCL, and LCL were 4%, 4%, and 5%, respectively (p<0.05 vs the ACL, Fig 5). The ACL also had a significantly greater normalized CSA values at 0 and 3 months compared to the other three tissues, and compared to the PT at 4.5 months of age (p<0.05). No significant differences were found between the tissues during adolescence (6 and 18 months, p>0.05).

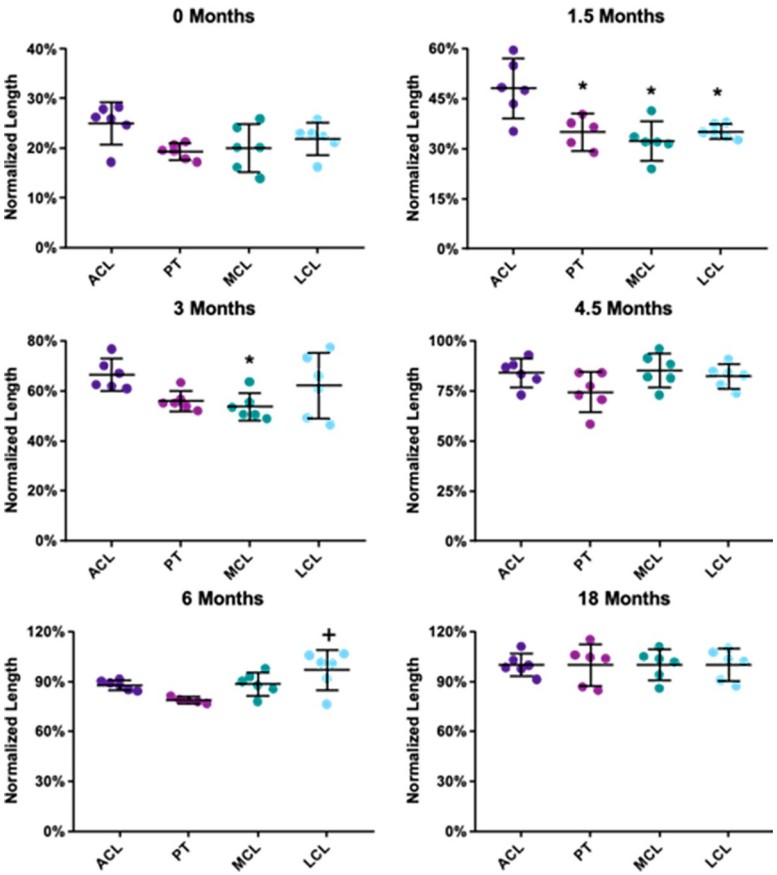

**Fig 4. Tissue length data compared across tissues at each age normalized to the late adolescent group.** Data for individual specimens are presented as points while mean and 95% confidence intervals are represented by dashes and lines. * denotes p<0.05 from ACL, + denotes p<0.05 from PT.

### Intra-tissue growth behavior

Statistical analyses of log-log plots between morphometric parameters (length and CSA) for each tissue (ACL, PT, MCL, LCL) revealed differences in the modality of growth for each tissue (Fig 6). The slope of best fit line for the CSA versus length plot of the ACL was 1.54, which was significantly different from the isometric slope of 2 (p<0.001) and favored allometric increases in length over increases in CSA. The slope of the CSA versus length plot for the PT was 1.85, which was not significantly different from the isometric slope (p = 0.25). Similarly, the slope of the MCL CSA versus length plot (1.84) did not differ from the isometric slope (p = 0.08). However, the slope of the LCL CSA versus length plot (1.72) was significantly different from that of an isometric slope (p = 0.002) and favored greater length change over CSA change.

### Inter-tissue differences in growth

Similar analyses were performed to compare log-log plots of morphologic growth across the four tissues of interest in the joint, with varied results depending on the parameter of interest (Fig 7). In terms of length, changes favored growth in the PT, MCL, and LCL over the ACL. Specifically, the slopes of the plots for the ACL relative to these tissues were 0.88 (p = 0.010), 0.82 (p<0.001), and 0.88 (p = 0.006), respectively. Length changes were not statistically different from an isometric slope in the PT versus MCL (slope = 1.02, p = 0.52), PT versus LCL (slope = 0.97, p = 0.39), or MCL versus LCL (slope = 1.03, p = 0.04) plots.

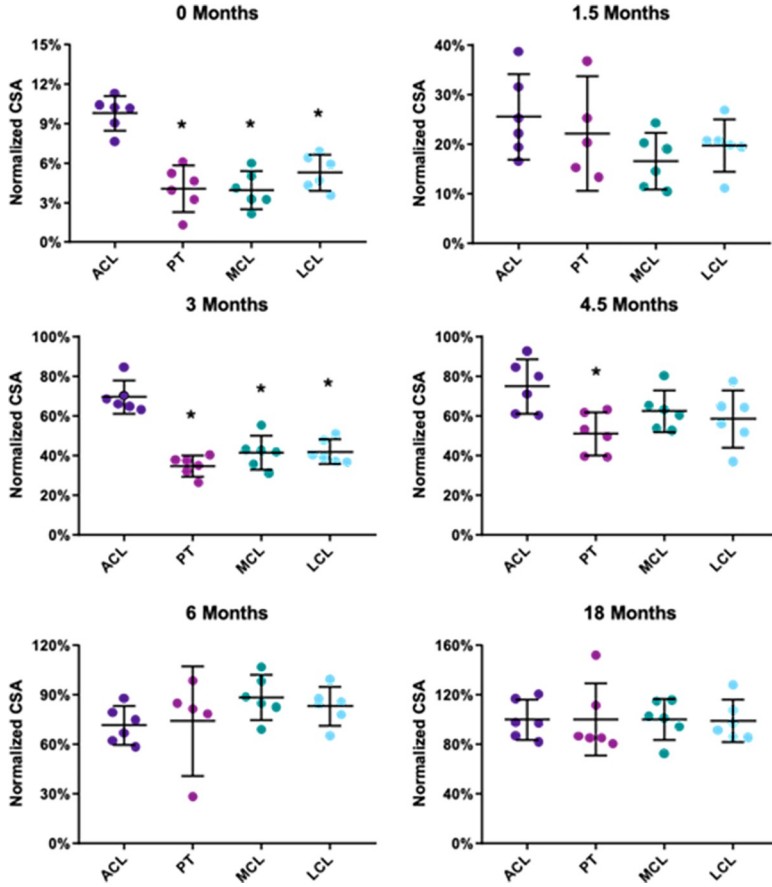

**Fig 5. Tissue CSA data compared across tissues at each age normalized to the late adolescent group.** CSA is significantly greater in the ACL compared to the other tissues (p<0.05) at ages including 0, 3, and 4.5 months. Data for individual specimens are presented as points while mean and 95% confidence intervals are represented by dashes and lines. * denotes p<0.05 from ACL.

Comparisons of the slopes of log-log plots for CSA growth revealed some similar changes (Fig 8). The ACL exhibited allometric behavior relative to all three of the other tissues with a slope of 0.68 versus the PT (p<0.001), 0.69 versus the MCL (p<0.001), and 0.78 versus the LCL (p<0.001). In all three of these cases, changes in CSA were greater for the MCL, LCL and PT relative to the ACL. CSA changes were not statistically different from an isometric slope in plots comparing the PT versus MCL (slope = 0.96, p = 0.11), PT versus LCL (slope = 0.85, p = 0.75), or the MCL versus LCL (slope = 1.10, p = 0.08) plots.

## Discussion

While previous studies have investigated growth across body segments, this work was performed to highlight differences in growth in tissues with similar structure in a single joint [17, 20, 24, 27, 28]. Here we presented data showing that all three ligaments and the patellar tendon studied in the knee joint increased markedly in size during growth. These changes included 4- to 5-fold increases in tissue lengths from birth through skeletal maturity alongside 10- to 20-fold increases in tissue CSA. However, changes in shape varied between tissues. Specifically, the ACL and LCL experienced allometric growth whereas the MCL and PT grew in an isometric manner. Additionally, while the increases in tissue length were similar across the tissues of

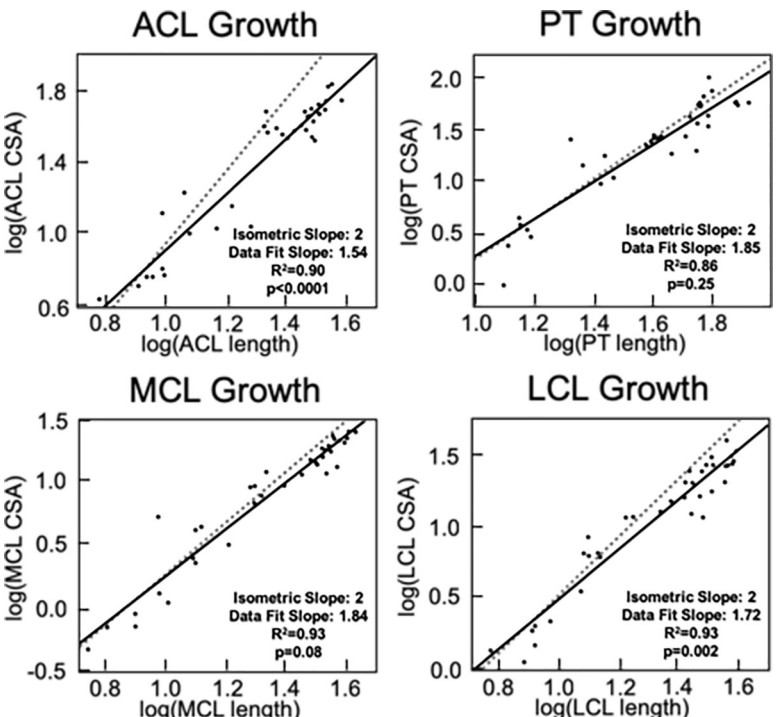

**Fig 6. Assessment of allometric growth within each tissue.** Comparisons of CSA and length for the ACL and LCL reveal allometric growth whereas the PT and MCL exhibit isometric growth. In these plots, the dashed line represents the line of isometry while the solid line represents the line of best fit for the data ($R^2$ values provided, $p<0.05$ denote statistical difference from isometric line, slopes denote standard and best fit lines).

interest, CSA changes varied between tissues as the percent change in ACL CSA was lower than in the other tissues.

The age-related size increases observed in our study match more limited data in the literature. Additionally, the 2-fold changes we found from juvenile to skeletally mature groups in MCL CSA between were similar in scale to those previously reported during the same time frame in a study of rabbit MCL size [29]. Related studies also reported more rapid growth in MCL CSA during the juvenile and early adolescent stages relative to later stages of growth, and our study reflected these findings as well [30]. In another study in rabbits, MCL length increased by approximately one third to one half during a 10-week period of juvenile growth [19]. Similarly, our findings suggested that the porcine MCL increased by just under one half of its length during a similar time frame. Our data build on these prior reports while allowing direct comparisons between tissues at a wider range of ages.

Our findings regarding the growth of the ACL suggest that there are age-specific changes in the geometric proportions of the tissue throughout skeletal growth in the pig model, and that the growth of the ACL does not parallel growth in the other tissues. Similarly, previous studies have shown that the CSA of the human ACL increases in size up to 10–12 years of age but halts in growth prior to the end of overall skeletal growth [8, 31]. Additional studies comparing the growth of the human ACL to muscles surrounding the knee have shown more rapid growth in the ACL halting prior to the end of muscle growth [32]. Our findings agree with these results demonstrating the allometric growth of the ACL relative to other tissues, while our findings add a more robust look into the timing of ACL growth in a relevant pre-clinical large animal model.

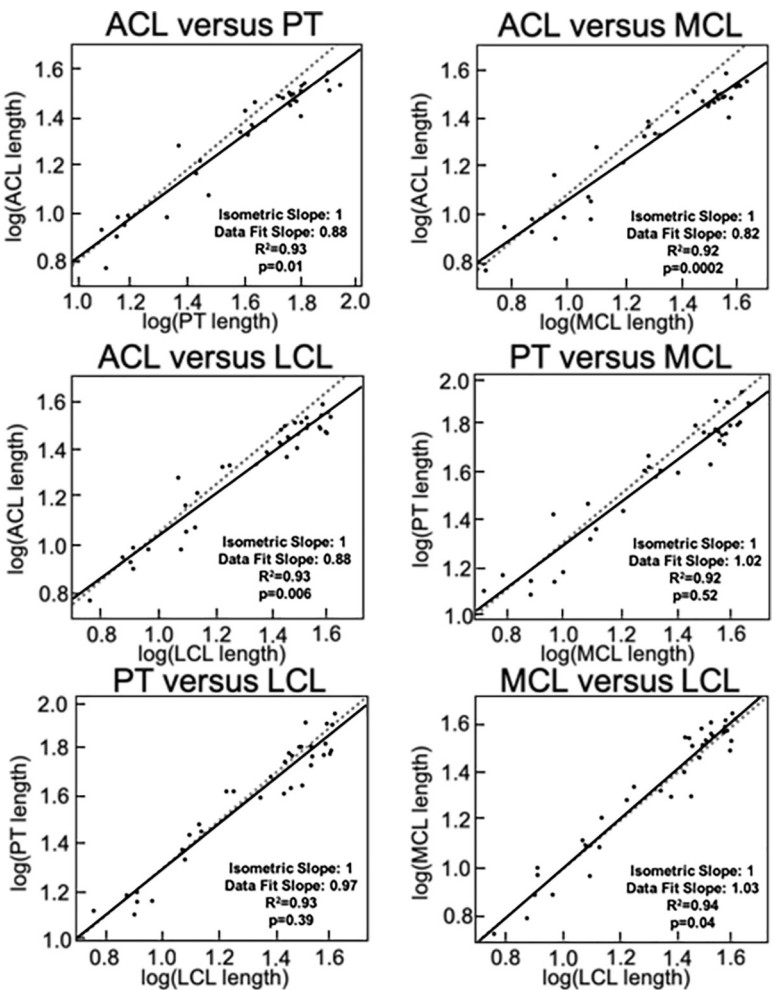

**Fig 7. Comparisons of tissue length reveal differing rates of growth between the tissues.** In these plots, the dashed line represents the line of isometry while the solid line represents the line of best fit for the data points ($R^2$ values provided, $p<0.05$ denote statistical difference from isometric line, slopes denote standard and best fit lines).

In order to extend these findings to the study of human growth, similar data should be analyzed in human subjects. MRI techniques have been previously employed to study the growth of musculoskeletal tissues in human populations, and some of the benefits and limitations of this approach have been described previously. MRI was used to study growth in the pediatric shoulder, where the ability to study changes in bone and soft tissues simultaneously was highlighted [33]. An additional MRI study reported age-related patterns in ligament anatomy including one reporting on the location of the femoral insertions of the collateral ligaments relative to the femoral growth plate [34]. Building on works such as these and the techniques described in our study, there is an opportunity to build on our understanding of ligament and tendon growth in human populations.

Our study also relates to previous work in the porcine model focused on growth and morphometric changes of bones, while adding data regarding several soft tissues in the knee. Other groups have reported age-related changes in the growth of long bones in the hind- and fore-limbs where they found differences in CSA and length change in the bones of the hind-limb, and variations in growth coefficients between the bones of the hind-limb versus the fore-limb [20]. Our findings agreed with this work suggesting that musculoskeletal tissues can undergo

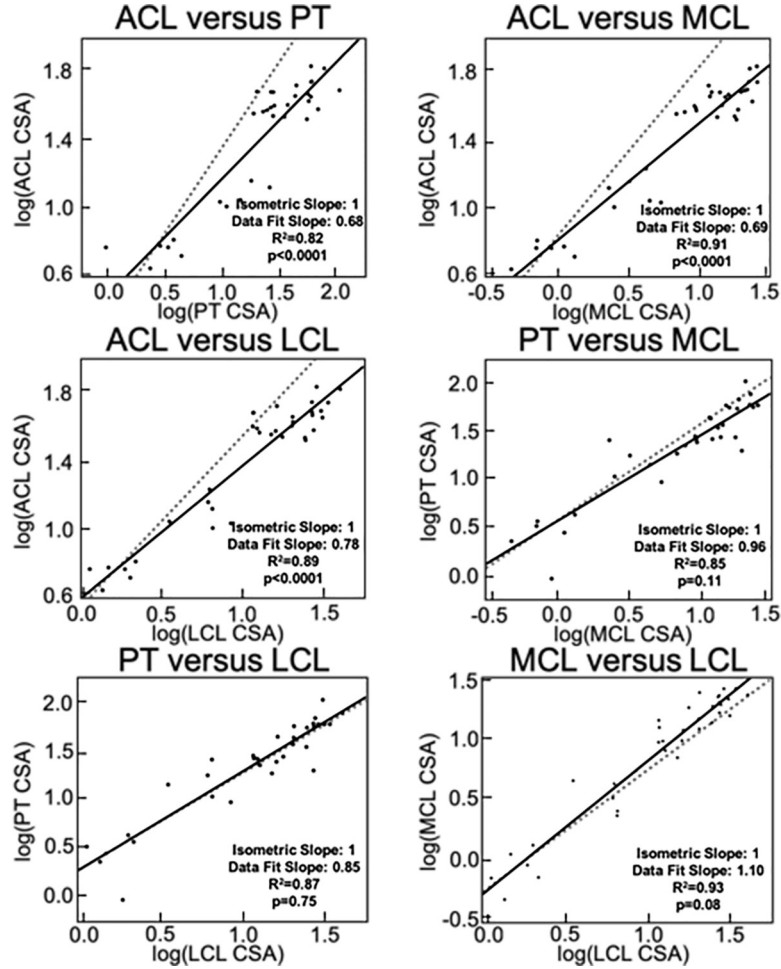

**Fig 8. Comparisons of tissue CSA show differing rates of growth between tissues, specifically between the ACL and the other three.** In these plots, the dashed line represents the mathematical line of isometry while the solid line represents the line of best fit for the data points ($R^2$ values provided, $p<0.05$ denote statistical difference from isometric line, slopes denote standard and best fit lines).

shape changes during growth and may occur on altered timelines relative to similar tissues. Furthermore, the porcine model has been used to study tissue mechanics during growth specifically for bone and cartilage [35–37]. Thus, the porcine model may also be a valuable tool to study ligament and tendon mechanics during growth.

The findings reported here can aide in designing clinical treatments for injured ligaments and tendons. For example, when developing reconstruction treatments for the ACL in growing patients, clinicians may need to be cognizant of age-specific morphology and remaining growth in the knee. The PT is a common graft for the ACL [38]; however, if the PT grows at different rates during the healing process or experiences different changes in shape this may not be as appropriate for certain age groups. In order to implement these findings in clinical treatments for human populations, this study should be repeated in a human population to confirm whether or not these findings are species-specific. Given the non-invasive nature of MRI studies, repeating this work with a wide range of ages during growth in a human cohort may reveal interesting differences in the relative size and shape of ligaments and tendons between species and age groups, as well as any interaction between these factors.

Furthermore, findings reported in this work specifically regarding allometric changes in the morphometry of the ACL near the onset of adolescence may be related to a growing clinical demographic of ACL injuries. Rates of ACL tears and subsequent reconstruction procedures have been rapidly increasing in skeletally immature patients over recent years, with some of the most rapid growth in injury incidence occurring in patients between 10–13 years of age [39, 40]. Unfortunately, these patients have relatively poor outcomes from ACL reconstruction, with much higher secondary ACL tear incidence rates compared to adult patients undergoing the same reconstruction treatments [41–45]. Improving our understanding of the age-specific size and shape of the ACL and how these parameters change with time may enable clinicians to tailor graft selection and reconstruction techniques to improve long-term outcomes in this young patient demographic.

These data also have implications in basic science research of ligaments and tendons. If developing a biomechanical model of the joint during growth, a single scaling factor cannot be applied across tissues within a single joint to create age-specific designs. Any biomechanical model of the knee should consider the corresponding relationships between length and CSA that are unique to a target age group. While outside of the scope of this project, analysis of the regional biomechanical function of these tendons and ligaments throughout growth may improve our understanding of the mechanisms and implications of these morphometric changes. When considering the consequences of these findings on the field of tissue engineering and applications for ligaments and tendons, tissue-specific growth should be taken into consideration when designing tissue engineered constructs for skeletally immature patients. The study of tissue-specific changes during growth could be applied to other species and tissues. This may reveal significant differences in tissue growth in precocial and altricial species, bipedal and quadrupedal species, and across upper and lower limbs within bipedal species.

Moving forward, we plan to replicate this study in male animals in order to investigate the sex-specific changes in soft tissue morphology within the knee. Previous studies showing significant differences in injury patterns between young male and female athletes suggest that the onset of adolescence leads to a disparity in ACL behavior between the sexes [46], and this future work may elucidate the impact of structural changes on these differences. In combination with findings on the specific tissue composition and effects of altered mechanical loading on soft tissue morphogenesis, the porcine model can be used to isolate the underlying mechanisms which inform soft tissue growth in post-natal development. Furthermore, we hope to analyze the biochemical composition of these tissues throughout growth to better understand the underlying changes during growth.

Limitations of this study include the cross-sectional experimental design, as all morphological changes between ages are based on data from separate animals. Future work using a longitudinal study design may be able to expand on the power of this work by eliminating inter-specimen variation. An additional consideration is that these findings may be breed specific within the porcine species. As such, it may be informative to repeat these measurements in other common porcine models, such as the Yucatan minipig. The use of two scanning systems (7.0T MRI and 9.4T MRI) may have resulted in some differences in the accuracy of direct comparisons between tissues from 0-month specimens and all other groups. Unfortunately, due to the small size of some tissues in the 0-month limbs (dimensions less than 7.0T voxel size) and the small bore size of the 9.4T scanner (less than the dimension of the 1.5-month joints) we were unable to perform a direct comparison of images taken from the tissues of interest across these scanners. Further limitations were introduced by the methodology for length calculations. Specifically, since the geometry of the insertion sites for the ligaments and tendons are complex (and partially out of the field of view in some cases), our length measurements were constrained to the midsubstance of the tissues. Additionally, the CSA measurements for each

tissue were gathered from the midsubstance and were averaged across the tissue. Thus, our measurements represent generalized metrics of the tissue substance. Quantification of region-specific variations in size during growth would be an interesting area for exploration.

While previous work has shown growth gradients across full limbs or body segments, this has shown that similar tissues within a single joint can grow at different rates. Our findings support previous literature suggesting that age-related morphometry changes vary between tissues in the body, while increasing our understanding of this phenomenon within comparisons between tissues within a single organ. This data has many potential implications in understanding musculoskeletal growth, human clinical applications, and emerging tissue engineering and regenerative medicine therapies.

## Supporting information

**S1 Table. Tissue length.**
(DOCX)

**S2 Table. Tissue cross-sectional area.**
(DOCX)

**S3 Table. Normalized tissue length.**
(DOCX)

**S4 Table. Normalized tissue cross-sectional area.**
(DOCX)

## Acknowledgments

The authors would like to thank Mr. Sean Simpson, the Swine Education Unit at North Carolina State University, and the Biomedical Research Imaging Center at the University of North Carolina–Chapel Hill for their contributions to this work.

## Author Contributions

**Conceptualization:** Stephanie G. Cone, Jeffrey T. Spang, Matthew B. Fisher.

**Data curation:** Stephanie G. Cone, Lynn A. Fordham, Matthew B. Fisher.

**Formal analysis:** Stephanie G. Cone, Hongyu Ru, Matthew B. Fisher.

**Funding acquisition:** Jorge A. Piedrahita, Jeffrey T. Spang, Lynn A. Fordham.

**Investigation:** Stephanie G. Cone, Hope E. Piercy, Emily P. Lambeth.

**Methodology:** Stephanie G. Cone, Hope E. Piercy, Emily P. Lambeth, Matthew B. Fisher.

**Project administration:** Matthew B. Fisher.

**Validation:** Stephanie G. Cone.

**Visualization:** Stephanie G. Cone.

**Writing – original draft:** Stephanie G. Cone.

**Writing – review & editing:** Stephanie G. Cone, Hope E. Piercy, Emily P. Lambeth, Hongyu Ru, Jorge A. Piedrahita, Jeffrey T. Spang, Lynn A. Fordham, Matthew B. Fisher.

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
