## [Decision Letter · Decision Letter 0]

13 Aug 2019

PONE-D-19-17174

Tissue-specific changes in size and shape of the ligaments and tendons of the porcine knee during post-natal growth

PLOS ONE

Dear Dr. Fisher,

Thank you for submitting your manuscript to PLOS ONE. After careful consideration, we feel that it has merit but does not fully meet PLOS ONE’s publication criteria as it currently stands. Therefore, we invite you to submit a revised version of the manuscript that addresses the points raised during the review process.

Please address the comments from the reviewers.

We would appreciate receiving your revised manuscript by Sep 27 2019 11:59PM. To enhance the reproducibility of your results, we recommend that if applicable you deposit your laboratory protocols in protocols.io, where a protocol can be assigned its own identifier (DOI) such that it can be cited independently in the future. For instructions see: http://journals.plos.org/plosone/s/submission-guidelines#loc-laboratory-protocols

We look forward to receiving your revised manuscript.

Kind regards,

Alejandro Almarza, Ph.D.

Academic Editor

PLOS ONE

Journal Requirements:

2. At this time, we request that you  please report additional details in your Methods section regarding animal care, as per our editorial guidelines: 1) Please provide details of animal welfare (e.g., shelter, food, water, environmental enrichment) 2) please describe any steps taken to minimize animal suffering and distress, such as by administering analgesics, and 3) please include the method of sacrifice. Thank you for your attention to these requests.

Reviewers' comments:

Reviewer's Responses to Questions

**Comments to the Author**

1. Is the manuscript technically sound, and do the data support the conclusions?

Reviewer #1: Yes

Reviewer #2: Yes

2. Has the statistical analysis been performed appropriately and rigorously? 

Reviewer #1: Yes

Reviewer #2: Yes

3. Have the authors made all data underlying the findings in their manuscript fully available?

Reviewer #1: Yes

Reviewer #2: Yes

4. Is the manuscript presented in an intelligible fashion and written in standard English?

Reviewer #1: Yes

Reviewer #2: Yes

5. Review Comments to the Author

Reviewer #1: This is an interesting topic and important work for the field of pediatrics and orthopedics.

The second paragraph in the introduction reads a bit abstract. For example, the "pioneering work by D'Arcy Thompson" can be revised, as the comparisons made in the present paper are measurements of iso/allometric scaling, which was pioneered by Huxley. (e.g., citation for definitions of isometry/allometry should be: "Terminology of Relative Growth." J. S. Huxley & G. Teissier, Nature 137: 780–781 (1936)). The Thompson citation (12) also needs revising because of typos.

How might the resolution differences between the 7.0T and 9.4T MRI scanner influence the quantitative outcomes for length/CSA? Validation comparisons of one time point (e.g., 1.5mo or younger) scanned using both 7.0T and 9.4T or discussion here would be helpful to explain potential variations.

What variations in animal size (weight) were found and how might this be related to iso/allometric changes in ligament size and length?

The rapid change in size/length that occurs from 1.5 to 3mo (e.g., ACL CSA) is interesting; do the tissue properties change during growth and could this influence the quantification of ligament size/length using MR? What methods were used to confirm that the MR images were accurately representing the growth of the soft tissues (e.g., dissection and measurement using calipers; assumptions based on morphology/imaging, etc)?

Reviewer #2: Summary:

The purpose of this work was to characterize temporal and tissue-specific (allometric vs. isometric) changes in porcine ACL, MCL, LCL, and PT between 0-18 months of age in a cross-sectional study design. Tissues were segmented from MR images, and 3D mesh models were constructed. Tissue length and mid-substance cross-sectional area (CSA) were determined from the segmentation models. Ligament and PT growth was generally allometric with greater elongation than increases in CSA; this trend was most evident in ACL growth. The most rapid growth occurred from 1.5-4.5 months of age. The authors propose that this knowledge, when coupled with age-specific investigations of biochemical and mechanical tissue properties, have broad applicability to the fields of tissue engineering, computational modeling, regenerative medicine, and surgical reconstruction.

General Comments:

This was a concise study with particular relevance to the porcine ACL model. The manuscript and figures were clear, save for a few areas suggested under the “Specific Comments”, below.

One area of relevance of this work that could be better emphasized and discussed is the clinical relevance of high rate of ACL rupture during late adolescence. It would seem that the allometric changes were most dramatic in the ACL compared to the other extra-articular tissues examined. Could the authors speculate in the Discussion (along with any supporting evidence) why this may be, and whether they think it has any relevance to ACL injury risk? That is, dig in to the details of the statements in the paragraph spanning lines 360-370. This shift in emphasis relates most directly to the results presented, and would make the discussion on the implications of the results to other tissues, joints, and species secondary to the actual data presented.

Although the final data are convincing, it would be valuable to state whether all segmentations were performed by a single segmenter, and what the reproducibility of the masks (or model generation) was. That is, how good was intra- and/or inter-segmenter agreement?

A general comment on the Results: a measure of data spread (either the standard deviation or 95% confidence intervals) should accompany the reporting of average values (e.g., average values throughout the first paragraph of ther Results).

Review the formatting of the References – there are a few typos in the numbering.

Specific Comments:

Line 43: Suggest revising to “…morphometric and mechanical properties enabling force transmission and movement.” since ligaments and tendons do not generate force, per se – muscles do.

Line 50: The sentence ending with “… as well as mechanical loading.” requires a reference.

Line 51: The first statement in the sentence on this line requires a reference: “Pioneering work by D’Arcy Thompson (12) and many others (ref(s)),…”

Lines 54-55: Suggest revising to “…and “allometry” describe changes in which the growth of a part do, or do not, match the growth…”

Specimen collection Lines 88-100: if known, please add the average mass +/- standard deviation of the animals comprising each age group.

Line 153: I assume that the log transformations on the data were done to account for the non-linear growth characteristics, but the rationale should be stated explicitly.

Line 154: The application of “established” here makes me think that the experimental data are being compared to data “established” in the literature, but I’m not sure that’s the intent – I think the authors are referring to the empirical slopes described in Figure 2B (?).

Related to the point above, it’s unclear where the isometric slope of 2 is derived from for the isometric slope of Length vs CSA. By definition, a slope of two is not isometric (i.e. not 1). Please provide a clearer explanation in the Methods.

Moving Lines 199-202 and 213-214 from the Results to the Statistical analysis section of the Methods would provide a clearer rationale and thorough explanation of the normalization step.

Line 282: Suggest revising statement to “… presented data that all three ligaments and the patellar tendon studied…” since data on the PCL were not presented, and only a single tendon was studied. Also, was there a rationale for not including the PCL in the analyses?

Lines 290-291: This opening sentence requires citations.

Lines 389-390: Suggest removing the “…comparisons between body segments and multiple joints” from the key conclusion statement since the study did not present these data.

6. PLOS authors have the option to publish the peer review history of their article (what does this mean?). If published, this will include your full peer review and any attached files.

Reviewer #1: No

Reviewer #2: No

---

## [Author Response · Author response to Decision Letter 0]

26 Sep 2019

Reviewer #1: 

This is an interesting topic and important work for the field of pediatrics and orthopedics.

We thank the reviewer for their kind words, and appreciate the feedback below.

The second paragraph in the introduction reads a bit abstract. For example, the "pioneering work by D'Arcy Thompson" can be revised, as the comparisons made in the present paper are measurements of iso/allometric scaling, which was pioneered by Huxley. (e.g., citation for definitions of isometry/allometry should be: "Terminology of Relative Growth." J. S. Huxley & G. Teissier, Nature 137: 780–781 (1936)). The Thompson citation (12) also needs revising because of typos.

We have revised the Thompson citation and adjusted the second paragraph as suggested, with the primary revisions reading as follows (Page 3, Lines 51-55):

 “Previous work by D’Arcy Thompson (13), Julian Huxley (14), and many others, have reported changes in the size and shape of biological tissues, resulting in the establishment of many terms and methods for classifying objects during growth. In this work, we focus on the terms isometry and allometry as defined by Huxley and how they can be used to describe morphological changes in tissues during skeletal growth. The terms isometry and allometry describe changes in which the growth of a part, do or do not, match the growth of the whole, respectively (13-15).”

How might the resolution differences between the 7.0T and 9.4T MRI scanner influence the quantitative outcomes for length/CSA? Validation comparisons of one time point (e.g., 1.5mo or younger) scanned using both 7.0T and 9.4T or discussion here would be helpful to explain potential variations.

Unfortunately we are unable to scan any joints from 1.5 months or older using the 9.4T methods (due to bore size) and some features of the 0 month joints would not be detectable following the 7T methods (ligaments with thicknesses of <2 voxels at the 7T resolution). Although we are not able to add experimental comparisons of the two methods, we have adapted the limitations section of the discussion to better address this issue. Text additions as follows (Page 19, Lines 430-436):

“The use of two scanning systems (7.0T MRI and 9.4T MRI) may have resulted in some differences in the accuracy of direct comparisons between tissues from 0-month specimens and all other groups. Unfortunately, due to the small size of some tissues in the 0-month limbs (dimensions less than 7.0T voxel size) and the small bore size of the 9.4T scanner (less than the dimension of the 1.5-month joints) we were unable to perform a direct comparison of images taken from the tissues of interest across these scanners.”

What variations in animal size (weight) were found and how might this be related to iso/allometric changes in ligament size and length?

Animal weight was not available for this cohort due to limitations at site of animal housing, although it would be an interesting metric to consider in future studies. Given the rapid growth of these animals, it is unlikely that variations in size would impact conclusions drawn between age groups.

The rapid change in size/length that occurs from 1.5 to 3mo (e.g., ACL CSA) is interesting; do the tissue properties change during growth and could this influence the quantification of ligament size/length using MR? What methods were used to confirm that the MR images were accurately representing the growth of the soft tissues (e.g., dissection and measurement using calipers; assumptions based on morphology/imaging, etc)?

Although changes in tissue properties may influence the intensity values associated with images at different ages, the process described in this manuscript only relies on the ability to distinguish the ligament tissue relative to surrounding tissues, which was clear at all ages. This being said, our group has previously assessed the accuracy of soft tissue shape in the juvenile porcine joint following the same 7T imaging and image processing methods described in this work by comparing to laser scan data of the same tissues. This represents small tissues at our lowest resolution. These comparisons were limited to tissues which hold their shape ex vivo, so this was done for the knee meniscus. We did not perform the study on ligament or tendon tissue which would deform substantially once the joint environment was disturbed. Nevertheless, we found that our MRI reconstructions resulted in similar geometries as the laser scans with root mean square error values of 0.6±0.1mm for overall surface points between methods. This is on the order of 1-2 voxels for our 7T scans. For our smallest ligament dimensions via 7T MRI in the current study were 20 and 27 voxels for CSA and length, respectively. Thus, we would expect the upper bound of the error of our measurements to be ~10% error, and for most larger specimens, we would expect the error to be much lower. Text reflecting these measurements has been added as follows (Page 7, Lines 139-145):

“Comparison studies of the 3D reconstructions of in situ menisci calculated from these MRI scans to a gold-standard surface reconstruction based on 3D laser scans (FARO Edge ScanArm ES, Lake Mary, FL) indicated a root mean square error (RMSE) of 0.57±0.1mm for points along the object surfaces and an RMSE of 0.66±0.2mm for the centroid points of menisci. Any primary sources of error translating from MRI-based measurements to physical tissue morphometries should be systemic, resulting in little impact on our comparisons between age groups.”

Reviewer #2

Summary: The purpose of this work was to characterize temporal and tissue-specific (allometric vs. isometric) changes in porcine ACL, MCL, LCL, and PT between 0-18 months of age in a cross-sectional study design. Tissues were segmented from MR images, and 3D mesh models were constructed. Tissue length and mid-substance cross-sectional area (CSA) were determined from the segmentation models. Ligament and PT growth was generally allometric with greater elongation than increases in CSA; this trend was most evident in ACL growth. The most rapid growth occurred from 1.5-4.5 months of age. The authors propose that this knowledge, when coupled with age-specific investigations of biochemical and mechanical tissue properties, have broad applicability to the fields of tissue engineering, computational modeling, regenerative medicine, and surgical reconstruction.

General Comments:

This was a concise study with particular relevance to the porcine ACL model. The manuscript and figures were clear, save for a few areas suggested under the “Specific Comments”, below.

Thank you for these comments and your helpful suggestions below, we have updated the manuscript accordingly.

One area of relevance of this work that could be better emphasized and discussed is the clinical relevance of high rate of ACL rupture during late adolescence. It would seem that the allometric changes were most dramatic in the ACL compared to the other extra-articular tissues examined. Could the authors speculate in the Discussion (along with any supporting evidence) why this may be, and whether they think it has any relevance to ACL injury risk? That is, dig in to the details of the statements in the paragraph spanning lines 360-370. This shift in emphasis relates most directly to the results presented, and would make the discussion on the implications of the results to other tissues, joints, and species secondary to the actual data presented.

Thank you for bringing this up. The changes in the ACL certainly could have implications in this important clinical problem, and we have incorporated further discussion of the topic in the suggested section of the manuscript (Page 17, Lines 387-397):

 Furthermore, findings reported in this work specifically regarding allometric changes in the morphometry of the ACL near the onset of adolescence may be related to a growing clinical demographic of ACL injuries. Rates of ACL tears and subsequent reconstruction procedures have been rapidly increasing in skeletally immature patients over recent years, with some of the most rapid growth in injury incidence occurring in patients between 10-13 years of age (39, 40). Unfortunately, these patients have relatively poor outcomes from ACL reconstruction, with much higher secondary ACL tear incidence rates compared to adult patients undergoing the same reconstruction treatments (41-45). Improving our understanding of the age- specific size and shape of the ACL and how these parameters change with time may enable clinicians to tailor graft selection and reconstruction techniques to improve long-term outcomes in this young patient demographic.

Although the final data are convincing, it would be valuable to state whether all segmentations were performed by a single segmenter, and what the reproducibility of the masks (or model generation) was. That is, how good was intra- and/or inter-segmenter agreement?

We have added the suggested information to the methods section as follows (Pages 6-7, Lines 132-136):

 “All models and measurements were generated and performed by a single author (SC), and repeatability for this process for these ligaments and tendons has been established to be highly repeatable, with intrauser correlation tests resulting in data fitting with R2 values of 0.97-0.98 and interuser correlation tests resulting in data fitting with R2 values of 0.84-0.99 across parameters.”

A general comment on the Results: a measure of data spread (either the standard deviation or 95% confidence intervals) should accompany the reporting of average values (e.g., average values throughout the first paragraph of their Results).

Thank you for pointing this out, we have added standard deviations throughout the results section, and highlighted that the 95% confidence intervals are available in supplemental information (Pages 9-10, Lines 207-209):

 “Throughout the results section, data are presented as mean ± standard deviation, and 95% confidence intervals are available in the Supplemental Material.”

Review the formatting of the References – there are a few typos in the numbering.

We have updated the references to account for these issues.

Specific Comments:

Line 43: Suggest revising to “…morphometric and mechanical properties enabling force transmission and movement.” since ligaments and tendons do not generate force, per se – muscles do.

Text has been edited as follows (Page 3, Line 43):

 “...morphometric and mechanical properties enabling force transmission and movement.”

Line 50: The sentence ending with “… as well as mechanical loading.” requires a reference.

Reference added as follows (Page 3, Line 50):

 “...stimuli including biochemical and cell signaling as well as mechanical loading (12).”

Line 51: The first statement in the sentence on this line requires a reference: “Pioneering work by D’Arcy Thompson (12) and many others (ref(s)),…”

Reference added as follows (Page 3, Line 53):

 “...classifying objects during growth (13-15).”

Lines 54-55: Suggest revising to “…and “allometry” describe changes in which the growth of a part do, or do not, match the growth…”

Text has been edited as follows (Page 3, Line 56):

 “...and “allometry” describe changes in which the growth of a part, do or do not, match the growth...”

Specimen collection Lines 88-100: if known, please add the average mass +/- standard deviation of the animals comprising each age group.

Unfortunately, animal weight is not available for this study, please see the response to Comment 3 from Reviewer 1.

Line 153: I assume that the log transformations on the data were done to account for the non-linear growth characteristics, but the rationale should be stated explicitly.

We have added explanations of the rationale to the methods section at the first mention of log transformations as follows (Page 8, Lines 179-180):

“In order to account for non-linear tissue growth characteristics, log-log plots (log10) were created...”

Line 154: The application of “established” here makes me think that the experimental data are being compared to data “established” in the literature, but I’m not sure that’s the intent – I think the authors are referring to the empirical slopes described in Figure 2B (?).

This wording has been updated to exclude the term “established” for clarity (Page 8, Line 180):

 “...comparing experimental data to established isometric slopes...”

Related to the point above, it’s unclear where the isometric slope of 2 is derived from for the isometric slope of Length vs CSA. By definition, a slope of two is not isometric (i.e. not 1). Please provide a clearer explanation in the Methods.

Additional text has been added clarifying that a slope of two represents isometric growth (as it is comparing the log of a mm^2 value to the log of a mm^1 value) instead of being a direct isometric slope (which would be 1). This relationship is commonly leveraged in bone growth studies to compare between parameters with different dimensions (Page 9, Lines 186-188):

“Isometric slopes of 1 (length vs length, CSA vs CSA) or 2 (CSA vs length) were used to account for differences in dimensions between mm and mm2 (17).”

Moving Lines 199-202 and 213-214 from the Results to the Statistical analysis section of the Methods would provide a clearer rationale and thorough explanation of the normalization step.

We have moved the suggested text (shown below) to the methods section as advised (Page 7-8, Lines 156-161, 167-168):

“To compare across tissues within specific ages, values for length were normalized to the average value at 18 months for each tissue. For example, at birth, the average ACL length was 25% of the average ACL length at 18 months...Values for CSA were also normalized to the average 18 month value for each tissue.”

Line 282: Suggest revising statement to “… presented data that all three ligaments and the patellar tendon studied…” since data on the PCL were not presented, and only a single tendon was studied. Also, was there a rationale for not including the PCL in the analyses?

We have revised the text as follows (Page 14, Lines 321-322):

“...presented data showing that all three ligaments and the patellar tendon studied...”

Regarding the PCL, we did not include this tissue in our analysis because of the significant, and sharp, change in angle in the PCL along its length (creating anterior-posterior and superior-inferior sections). Thus, we only compared tissues with a single primary axis.

Lines 290-291: This opening sentence requires citations.

References have been added as follows (Page 14, Line 321):

“...this work was performed to highlight differences in growth in tissues with similar structure in a single joint (17, 20, 24, 27, 28).”

Lines 389-390: Suggest removing the “…comparisons between body segments and multiple joints” from the key conclusion statement since the study did not present these data.

We have edited the manuscript to exclude this phrase as suggested and the statement now reads as follows (Page 20, Lines 447-448):

“...increasing our understanding of this phenomenon within comparisons between tissues within a single organ...”

---

## [Editor Report · Decision Letter 1]

2 Oct 2019

Tissue-specific changes in size and shape of the ligaments and tendons of the porcine knee during post-natal growth

PONE-D-19-17174R1

Dear Dr. Fisher,

We are pleased to inform you that your manuscript has been judged scientifically suitable for publication and will be formally accepted for publication once it complies with all outstanding technical requirements.

With kind regards,

Alejandro Almarza, Ph.D.

Academic Editor

PLOS ONE

Additional Editor Comments (optional):

Nice work.

---

## [Editor Report · Acceptance letter]

15 Oct 2019

PONE-D-19-17174R1 

Tissue-specific changes in size and shape of the ligaments and tendons of the porcine knee during post-natal growth 

Dear Dr. Fisher:

I am pleased to inform you that your manuscript has been deemed suitable for publication in PLOS ONE. Congratulations! Your manuscript is now with our production department. 

With kind regards,

on behalf of

Dr. Alejandro Almarza 

Academic Editor

PLOS ONE